# On Generalized Schürmann Entropy Estimators

**DOI:** 10.3390/e24050680

**Published:** 2022-05-11

**Authors:** Peter Grassberger

**Affiliations:** Jülich Supercomputing Center, Jülich Research Center, D-52425 Jülich, Germany; p.grassberger@fz-juelich.de

**Keywords:** entropy estimates, mutual information estimates, undersampling, Bayesian, bias, variance

## Abstract

We present a new class of estimators of Shannon entropy for severely undersampled discrete distributions. It is based on a generalization of an estimator proposed by T. Schürmann, which itself is a generalization of an estimator proposed by myself.For a special set of parameters, they are completely free of bias and have a finite variance, something which is widely believed to be impossible. We present also detailed numerical tests, where we compare them with other recent estimators and with exact results, and point out a clash with Bayesian estimators for mutual information.

## 1. Introduction

It is well known that estimating (Shannon) entropies from finite samples is not trivial. If one naively replaces the probability $p_{i}$ to be in “box” *i* by the observed frequency, $p_{i}\approx n_{i}/N$, statistical fluctuations tend to make the distribution look less uniform, which leads to an underestimation of the entropy. There have been numerous proposals on how to estimate and eliminate this bias [1,2,3,4,5,6,7,8,9,10,11,12,13,14,15,16,17,18,19,20,21,22]. Some make quite strong assumptions [5,7]; others use Bayesian methods [6,11,12,19,21,22]. As pointed out in [4,13,14,17], one can devise estimators with arbitrarily small bias (for sufficiently large *N* and fixed $p_{i}$), but these will then have very large statistical errors. As conjectured in [4,13,14,15,17], the variance of any estimator whose bias vanishes will have a diverging variance.

Another widespread belief is that Bayesian entropy estimators cannot be outperformed by non-Bayesian ones for severely undersampled cases. The problem with Bayesian estimators is of course that they depend on a good choice of prior distributions, which is not always easy, and they tend to be slow. One positive feature of a non-Bayesian estimator proposed in [14] is that it is extremely fast since it works precisely like the ‘naive’ (or maximum-likelihood) estimator, except that the logarithms used there are replaced by a function $G_{n}$ defined on integers, which can be precomputed by means of a simple recursion. While the estimator of [14] seems in general to be a reasonable compromise between bias and variance, it was shown in [15] that it can be improved—as far as bias is concerned, at the cost of increased variance—by generalizing $G_{n}$ to a one-parameter family of functions $G_{n}\left(a\right)$.

In the present paper, we show that the Grassberger–Schürmann approach [14,15] can be further improved by using different functions $G_{n}\left(a_{i}\right)$ for each different realization *i* of the random variable. Indeed, the $a_{i}$ can be chosen such that the estimator is completely free of bias and yet has a finite variance—although, to be honest, the optimal parameters $a_{i}$ can be found only if the exact distribution is known (in which case also the entropy can be computed exactly). We show that—even if the exact, optimal $a_{i}$ is not known—the new estimator can reduce the bias very much, without inducing unduly large variances, provided the distribution is sufficiently much undersampled.

We test the proposed estimator numerically with simple examples, where we produce bias-free entropy estimates, e.g., from pairs of ternary variables, something which, to my knowledge, is not possible with any Bayesian method. We also use it for estimating mutual information (MI) in cases where one of the two variables is binary, and the other one can take very many values. In the limit of severe undersampling and of no obvious regularity in the true probabilities, MI cannot be estimated unambiguously. In that limit, the present algorithm seems to choose systematically a different outcome from Bayesian methods for reasons that are not yet clear.

## 2. Basic Formalism

In the following, we use the notation of [14]. As in this reference, we consider M>1 “boxes” (states, possible experimental outcomes, etc.) and N>1 points (samples, events, and particles) distributed randomly and independently into the boxes. We assume that each box has weight $p_{i}$ (i=1,…M) with $\sum_{i}p_{i}=1$. Thus each box *i* will contain a random number $n_{i}$ of points, with $E\left[n_{i}\right]=p_{i}N$. Their joint distribution is multinomial,
(1)$$P\left(n_{1},n_{2},\ldots n_{M};N\right)=N!\prod_{i=1}^{M}\frac{p_{i}^{n_{i}}}{n_{i}!},$$
while the marginal distribution in box *i* is binomial,
(2)P(ni;pi,N)=Nnipini(1−pi)N−ni.

Our aim is to estimate the entropy,
(3)$$H=-\sum_{i=1}^{M}p_{i}\ln p_{i}=\ln N-\frac{1}{N}\sum_{i=1}^{M}z_{i}\ln z_{i},$$
with $z_{i}\equiv E\left[n_{i}\right]=p_{i}N$, from an observation of the numbers $\left\{n_{i}\right\}$ (in the following, all entropies are measured in “natural units”, not in bits). The estimator $\hat{H}\left(n_{1},\ldots n_{M}\right)$ will of course have both statistical errors and a bias, i.e., if we repeat this experiment, the average of $\hat{H}$ will, in general, not be equal to *H*,
(4)$$\Delta \left[\hat{H}\right]\equiv E\left[\hat{H}\right]-H\neq 0,$$
as will also be its variance $\mathrm{Var}[\hat{H}]$. Notice that for computing $E[\hat{H}]$, we need only Equation (Equation 2), not the full multinomial distribution of Equation (Equation 1). However, if we want to compute this variance, we additionally need the joint marginal distribution in two boxes,
(5)$$\begin{array}{ccc} P(n_{i},n_{j};p_{i},p_{j},N) & = & \frac{N!}{n_{i}!n_{j}!\left(N-n_{i}-n_{j}\right)!}\;\times \\ & & p_{i}^{n_{i}}p_{j}^{n_{j}}{\left(1-p_{i}-p_{j}\right)}^{N-n_{i}-n_{j}}, \end{array}$$
in order to compute the covariances between different boxes. Notice that these covariances were not taken into account in [13,17], whence the variance estimations in these papers are, at best, approximate.

In the following, we are mostly interested in the case where we are close to the limit N→∞,M→∞, with M/N (the average number of points per box) being finite and small. In this limit, the variance will go to zero (because essentially one averages over many boxes), but the bias will remain finite. The binomial distribution, Equation (Equation 2), can be replaced then by a Poisson distribution
(6)$$P_{\mathrm{Poisson}}\left(n_{i};z_{i}\right)=\frac{z_{i}^{n_{i}}}{n_{i}!}e^{-z_{i}}.$$

However, as we shall see, it is in general not good advice to jump right to this limit, even if we are close to it. More generally, we shall therefore also be interested in the case of large but finite *N*, where also the variance is positive, and we will discuss the balance between demanding minimal bias versus minimal variance.

Indeed it is well known that the *naive* (or ‘maximum-likelihood’) estimator, obtained by assuming $z_{i}=n_{i}$ without fluctuations,
(7)$${\hat{H}}_{\mathrm{naive}}=\ln N-\frac{1}{N}\sum_{i=1}^{M}n_{i}\ln n_{i},$$
is negatively biased, $\Delta {\hat{H}}_{\mathrm{naive}}<0$.

In order to estimate *H*, we have to estimate $p_{i}\ln p_{i}$ or equivalently $z_{i}\ln z_{i}$ for each *i*. Since the distribution of $n_{i}$ depends, according to Equation (Equation 2), on $z_{i}$ only, we can make the rather general ansatz [4,14] for the estimator
(8)$$\hat{z_{i}\ln z_{i}}=n_{i}\phi \left(n_{i}\right)$$
with a yet unknown function ϕ(n). Notice that $\hat{H}$ becomes with this ansatz a sum over strictly positive values of $n_{i}$. Effectively this means that we have assumed that observing an outcome $n_{i}=0$ does not give any information: if $n_{i}=0$, we do not know whether this is because of statistical fluctuations or because $p_{i}=0$ for that particular *i*.

The resulting entropy estimator is then [14]
(9)$${\hat{H}}_{\phi }=\ln N-\frac{M}{N}\bar{n\phi (n)}$$
with the overbar indicating an average over all boxes,
(10)$$\bar{n\phi (n)}=\frac{1}{M}\sum_{i=1}^{M}n_{i}\phi \left(n_{i}\right).$$

Its bias is
(11)$$\Delta H_{\phi }=\frac{M}{N}\left(\bar{z\ln z}-\bar{E_{N,z}\left[n\phi (n)\right]}\right).$$
with
(12)$$E_{N,z}\left[f_{n}\right]=\sum_{n=1}^{\infty }f_{n}P_{\mathrm{binom}}\left(n;p=z/N,N\right).$$
being the expectation value for a typical box (in the following we shall suppress the box index *i* to simplify notation, wherever this makes sense).

In the following, ψ(x)=dlnΓ(x)/dx is the digamma function, and
(13)$$E_{1}\left(x\right)=\Gamma \left(0,x\right)=\int_{1}^{\infty }\frac{e^{-xt}}{t}dt$$
is an exponential integral (Ref. [23], paragraph 5.1.4). It was shown in [14] that
(14)$$E_{N,z}\left[n\psi \left(n\right)\right]=z\ln z+z\left[\psi \left(N\right)-\ln N\right]+z\int_{0}^{1-z/N}\frac{x^{N-1}dx}{1-x}\;,$$
which simplifies in the Poisson limit (N→∞, *z* fixed) to
(15)$$E_{N,z}\left[n\psi \left(n\right)\right]\to z\ln z+zE_{1}\left(z\right)\;.$$

Equations (Equation 14) and (Equation 15) are the starting points of all further analysis. In [14], it was proposed to re-write Equation (Equation 15) as
(16)$$E_{N,z}\left[nG_{n}\right]\to z\ln z+zE_{1}\left(2z\right)\;,$$
where
(17)$$G_{n}=\psi \left(n\right)+{\left(-1\right)}^{n}\int_{0}^{1}\frac{x^{n-1}}{x+1}dx.$$
The advantages are that $G_{n}$ can be evaluated very easily by recursion (here γ=0.57721… is the Euler–Mascheroni constant), $G_{1}=G_{2}=-\gamma -\ln2,\;\;G_{2n+1}=G_{2n},\;$ and $G_{2n+2}=G_{2n}+\frac{2}{2n+1}$, and neglecting the second term, $zE_{1}\left(2z\right)$ gives an excellent approximation unless *z* is exceedingly small, i.e., unless the numbers of points per box are very small such that the distribution is very severely undersampled. Thus the entropy estimator proposed in [14] was simply
(18)$${\hat{H}}_{G}=\ln N-\frac{1}{N}\sum_{i=1}^{M}n_{i}G_{n_{i}}.\;\left(\mathrm{Poisson}\right)$$
Furthermore, since $zE_{1}\left(2z\right)$ is strictly positive, neglecting it gives a negative bias in ${\hat{H}}_{G}$, and one can show rigorously that this bias is smaller than that of [1,3].

## 3. Schürmann and Generalized Schürmann Estimators

The easiest way to understand the Schürmann class of estimators [15] is to define, instead of $G_{n}$, a one-parameter family of functions
(19)$$G_{n}\left(a\right)=\psi \left(n\right)+{\left(-1\right)}^{n}\int_{0}^{a}\frac{x^{n-1}}{x+1}dx.$$

Notice that $G_{n}\left(1\right)=G_{n}$ and $G_{n}\left(0\right)=\psi \left(n\right)$.

Let us first discuss the somewhat easier Poissonian limit, where
(20)$$\begin{array}{ccc} E_{N,z} & & \;\;\;\;\;\;\;[n\left(G_{n}\left(a\right)-\psi \left(n\right)\right)]=\\ & & =\sum_{n=1}^{\infty }{\left(-1\right)}^{n}nP_{\mathrm{Poisson}}\left(n,z\right)\int_{0}^{a}\frac{x^{n-1}}{x+1}dx\\ & & =-ze^{-z}\int_{0}^{a}\frac{dx}{x+1}e^{-xz}\\ & & =-z(E_{1}\left(z\right)-E_{1}\left(\left(1+a\right)z\right)), \end{array}$$
which gives
(21)$$E_{N,z}\left[nG_{n}\left(a\right)\right]=z\ln z+zE_{1}\left(\left(1+a\right)z\right).$$
Using—to achieve greater flexibility—different parameters $a_{i}$ for different boxes, and neglecting the second term in the last line of Equation (Equation 20), we obtain finally by using Equation (Equation 3)
(22)$${\hat{H}}_{\mathrm{Schuermann}}=\ln N-\frac{1}{N}\sum_{i=1}^{M}n_{i}G_{n_{i}}\left(a_{i}\right)\;\left(\mathrm{Poisson}\right).$$
Indeed, the last term in Equation (Equation 20) can always be neglected for sufficiently large *a* because $0<E_{1}\left(\left(1+a\right)z\right)<\exp\left(-\left(1+a\right)z\right)/\left(1+a\right)z$ for any real a>−1.

Equation (Equation 22) might suggest that using larger $a_{i}$ would always give an improvement because bias is reduced, but this would not take into account that larger $a_{i}$ might lead to larger variances. However, the optimal choices of the parameters $a_{i}$ are not obvious. Indeed, in spite of the ease of derivations in the Poissonian limit, it is much better to avoid it and to use the exact binomial expression.

For the general binomial case, the algebra is a bit more involved. By somewhat tedious but straightforward algebra, one finds that
(23)EN,z[n(Gn(a)−ψ(n))]==∑n=1∞(−1)nnNnpn(1−p)N−n∫0axn−1x+1dx=−pN∫0adxx+1∑n=1∞N−1n−1(−px)n−1(1−p)N−n=−pN∫0adxx+1(1−p−px)N−1=−z∫0adxx+1[1−(1+x)zN]N−1.
One immediately checks that this reduces, in the limit (N→∞, *z* fixed), to Equation (Equation 20). On the other hand, by substituting
(24)$$x\to t=1-\frac{(1+x)z}{N}$$
in the integral, we obtain
(25)$$E_{N,z}\left[nG_{n}\left(a\right)-\psi \left(n\right)\right)]=-z\int_{1-(1+a)z/N}^{1-z/N}\frac{t^{N-1}dt}{1-t}.$$
Finally, by combining with Equation (Equation 14), we find [15]
(26)$$\begin{array}{ccc} E_{N,z}[n\left(G_{n}\left(a\right)\right] & = & z\ln z+z[\psi (N)-\ln N]+\\ & + & z\int_{0}^{1-(1+a)z/N}\frac{x^{N-1}dx}{1-x} \end{array}$$
and, using again Equation (Equation 3),
(27)$${\hat{H}}_{\mathrm{opt}}=\psi \left(N\right)-\frac{1}{N}\sum_{i=1}^{M}n_{i}G_{n_{i}}\left(a_{i}\right),\;\left(\mathrm{binomial}\right)$$
with a correction term which is 1/N times a sum over the integrals in Equation (Equation 26). This correction term vanishes, if all integration ranges vanish. This happens when $1-\left(1+a_{i}\right)z_{i}/N=0$ for all *i*, or
(28)$$a_{i}=a_{i}^{*}\equiv \frac{1-p_{i}}{p_{i}}\;\forall i.$$
This is a remarkable result, as it shows that in principle, there exists always an estimator which has zero bias and yet finite variance. In [15], one single parameter *a* was used, which is why we call our method a generalized Schürmann estimator.

When all box weights are small, $p_{i}\ll 1$ for all *i*, then these bias-optimal values $a_{i}^{*}$ are very large. However, for two boxes with $p_{1}=p_{2}=1/2$, e.g., the bias vanishes already for $a_{1}=a_{2}=1$, i.e., for the estimator of Grassberger [14]!

In order to test the latter, we drew $10^{8}$ triplets of random bits (i.e., N=3, $p_{0}=p_{1}=1/2$), and estimated ${\hat{H}}_{\mathrm{naive}}$ and ${\hat{H}}_{G}$ for each triplet. From these, we computed averages and variances, with the results ${\hat{H}}_{\mathrm{naive}}=0.68867\left(4\right)$ bits and ${\hat{H}}_{G}=0.99995\left(4\right)$ bits. We should stress that the latter requires the precise form of Equation (Equation 27) to be used, with ψ(N) neither replaced by lnN nor by $G_{N}$.

Since there is no free lunch, there must of course be some problems in the limit when parameters $a_{i}$ are chosen to be nearly bias-optimal. One problem is that one cannot, in general, choose $a_{i}$ according to Equation (Equation 28), because the $p_{i}$ is unknown. In addition, it is in this limit (and more generally when $a_{i}>>1$) that variances blow up. In order to see this, we have to discuss in more detail the properties of the functions $G_{n}\left(a\right)$.

According to Equation (Equation 19), $G_{n}\left(a\right)$ is a sum of two terms, both of which can be computed, for all positive integer *n*, by recursion. The digamma function ψ(n) satisfies
(29)ψ(1)=−γ,ψ(n+1)=ψ(n)+1/n.
Let us denote the second term in Equation (Equation 19) as $g_{n}\left(a\right)$. It satisfies the recursion
(30)$$g_{1}\left(a\right)=-\ln\left(1+a\right),\;g_{n+1}\left(a\right)=g_{n}\left(a\right)-{\left(-a\right)}^{n}/n.$$
Thus, while ψ(n) is monotonic and slowly increasing, $g_{n}\left(a\right)$ has alternating sign and increases, for a>1, exponentially with *n*. As a consequence, also $G_{n}\left(a\right)$ is non-monotonic and diverges exponentially with *n*, whenever a>1. Therefore an estimator such as ${\hat{H}}_{\mathrm{opt}}$ gets, unless all $n_{i}$ are very small, increasingly large contributions of alternating signs. As a result, the variances will blow up, unless one is very careful to keep a balance between bias and variance.

To illustrate this, we drew tuples of independent and identically distributed binary variables $\left\{s_{1},\ldots s_{N}\right\}$ with $p_{0}=3/4$ and $p_{1}=1/4$. For $a_{0}$, we chose $a_{0}=a_{0}^{*}=1/3$ because this should minimize the bias and should not create problems with the variance. We should expect such problems, however, if we would take $a_{1}=a_{1}^{*}=3$, although this would reduce the bias to zero. Indeed we found for N=100 that the variance of the estimator exploded for all practical purposes as soon as $a_{1}>1.4$, while the results were optimal for $0.5<a_{1}\leq 1$ (bias and statistical error were both $<10^{-5}$ for $10^{8}$ tuples). On the other hand, for pairs (N=2), we had to use much larger values of $a_{1}$ for optimality, and $a_{1}=3$ gave indeed the best results (see Figure 1). A similar plot for ternary variables is shown in Figure 2, where we see again that *a*-values near the bias-optimal ones gave estimates with zero almost zero bias and acceptable variance for the most undersampled case N=2. Again, using the the exact bias-optimal values would have given unacceptably large variances for large *N*.

The message to be learned from this is that we should always keep all $a_{i}$ sufficiently small such that $a_{i}^{n_{i}}$ is not much larger than 1 for any of the observed values of $n_{i}$.

## 4. Estimating Mutual and Conditional Information

Finally, we apply our estimator to two problems of mutual information (MI) estimation discussed in [22] (actually, the problems were originally proposed by previous authors, but we shall compare our results mainly to those in [22]). In each of these problems, there are two discrete random variables: *X* has many (several thousand) possible values, while *Y* is binary. Moreover, the marginal distribution of *Y* is uniform, p(y=0)=p(y=1)=1/2, while the *X* distributions are highly non-uniform. Finally—and that is crucial—the joint distributions show no obvious regularities.

The MI is estimated as I(X:Y)=H(Y)−H(Y|X). Since H(Y)=1 bit, the problem essentially burns down to estimate the conditional probabilities p(y|x). The data are given in terms of a large number of independent and identically distributed sampled pairs (x,y) (250,000 pairs for problem I, called ‘PYM’ in the following, and 50,000 pairs for problem II, called ‘spherical’ in the following). The task is to draw random subsamples of size *N*, to estimate the MI from each subsample, and to calculate averages and statistical widths from these estimates.

Results are shown in Figure 3. For large *N*, our data agree perfectly with those in [22] and in the previous papers cited in [22]. However, while the MI estimates in these previous papers all increase with decreasing *N*, and those in [22] stay essential constant (as we would expect, since a good entropy estimator should not depend on *N*, and conditional entropies should decrease with *N* for not so good estimators), our estimated MI decreases to zero for small *N*.

This looks at first sight like a failure of our method, but it is not. As we said, the joint distributions show no regularities. For small *N* most values of *X* will show up at most once, and if we write the sequence of y–values in a typical tuple, it will look like a perfectly random binary string. The modeler knows that it actually is not random, because there are correlations between *X* and *Y*. However, no algorithm can know this, and any good algorithm should conclude that H(Y|X)=H(Y)=1 bit. Why, then, was this not found in the previous analyses? In all these, Bayesian estimators were used. If the priors used in these estimators were chosen in view of the special structures in the data (which are, as we should stress again, not visible from the data, as long as these are severely undersampled), then the algorithms can, of course, make use of these structures and avoid the conclusion that H(Y|X)=1 bit.

## 5. Conclusions

In conclusion, we gave an entropy estimator with zero bias and finite variance. It builds on an estimator by Schürmann [15], which itself is a generalization of [14]. It involves a real-valued parameter for each possible realization of the random variable, and bias is reduced to zero by choosing these parameters properly. However, this choice would require that we know already the distribution, which is of course not the case. Nevertheless we can reduce the bias very much for severely undersampled cases. In cases with moderate undersampling, choosing these zero-bias parameters would give very large variances and would thus be useless. Nevertheless, by judicious parameter choices, we can obtain extremely good entropy estimates. Finding good parameters is non-trivial, but is made less difficult by the fact that the method is very fast.

Finally, we pointed out that Bayesian methods, which have been very popular in this field, have the danger of choosing “too good” priors, i.e., choosing priors which are not justified by the data themselves and are thus misleading, although both the bias and the observed variances seem to be small.

I thank Thomas Schürmann for the numerous discussions, and Damián Hernández for both discussions and for sending me the data for Figure 3.

## Figures and Tables

**Figure 1 entropy-24-00680-f001:**
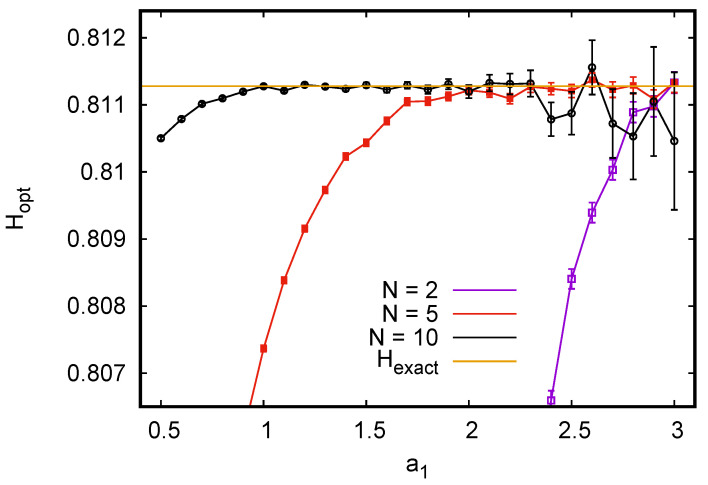
Estimated entropies (in bits) of *N*-tuples of independent and identically distributed random binary variables with $p_{0}=3/4$ and $p_{1}=1/4$, using the optimized estimator ${\hat{H}}_{\mathrm{opt}}$ defined in Equation (Equation 27). The parameter $a_{0}$ was kept fixed at its optimal value $a_{0}=1/3$, while $a_{1}$ is varied in view of possible problems with the variances, and is plotted on the horizontal axis. For each *N* and each value of $a_{1}$, $10^{8}$ tuples were drawn. The exact entropy for $p_{0}=3/4$ and $p_{1}=1/4$ is 0.811278… bits, and is indicated by the horizontal straight line.

**Figure 2 entropy-24-00680-f002:**
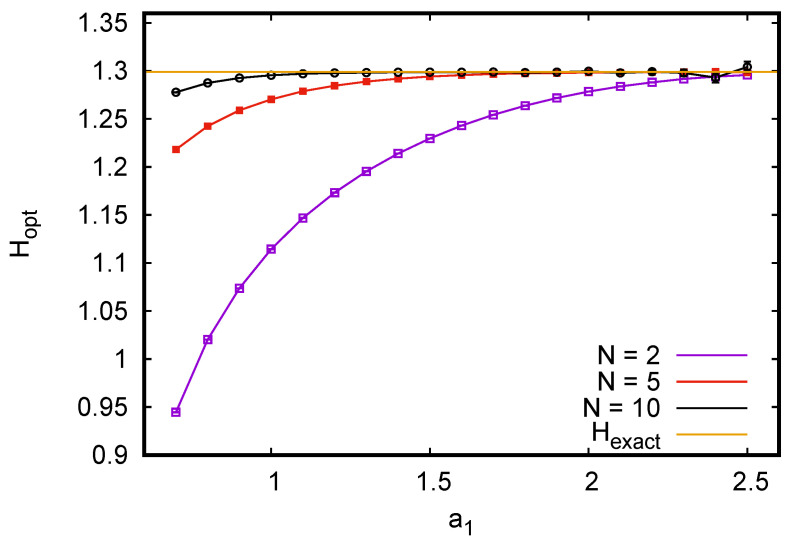
Estimated entropies (in bits) of *N*-tuples of independent and identically distributed random ternary variables with $p_{0}=0.625,$$p_{1}=0.25$, and $p_{2}=0.125$, using the optimized estimator ${\hat{H}}_{\mathrm{opt}}$ defined in Equation (Equation 27). The parameter $a_{0}$ was kept fixed at its optimal value $a_{0}^{*}=0.6$, while $a_{1}$ and $a_{2}$ varied in view of possible problems with the variances. More precisely, we used $a_{2}=1+4\left(a_{1}-1\right)$, so that the plot ends at the bias-free value $a_{2}^{*}=7.0$ and at a value of $a_{1}$ slightly smaller than $a_{1}^{*}=2.5$. For each *N* and each value of $a_{1}$, $10^{8}$ tuples were drawn. The exact entropy is 1.29879… bits, and is indicated by the horizontal straight line.

**Figure 3 entropy-24-00680-f003:**
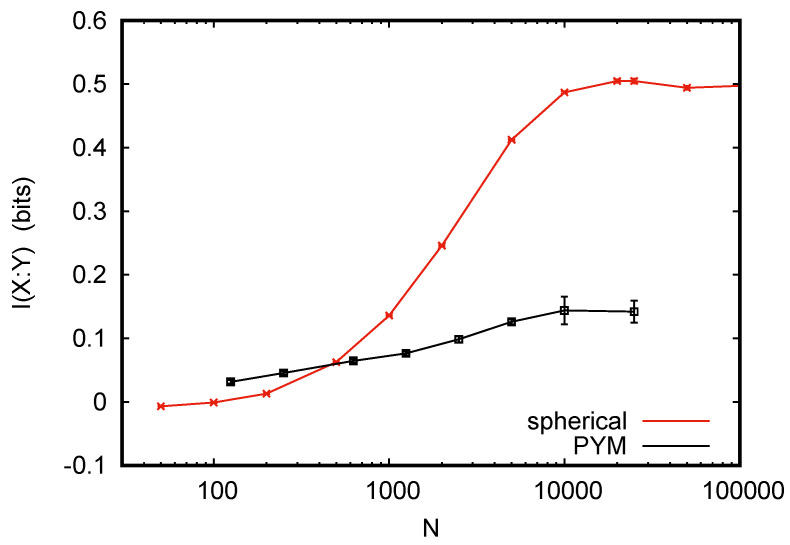
Estimated mutual information (in bits) of *N*-tuples of independent and identically distributed random subsamples from two distributions given in [22]. The data for “PYM”, originally due to [24], consist of 250,000 pairs (x,y) with binary *y* with p(y=0)=p(y=1)=1/2, and *x* being uniformly distributed over 4096 values. Thus each x–value is realized ≈60 times, and we classify them into 5 classes depending on the associated y–values: (i) very heavily biased toward y=1, (ii) moderately biased toward y=1, (iii) y–neutral, (iv) moderately biased toward y=0, and (v) heavily biased toward y=0. When we estimated conditional entropies H(Y|X) for randomly drawn subsamples, we kept this classification and choose $a_{y}$ accordingly: For class (iii) we used $a_{0}=a_{1}=1$, for class (ii) we used $a_{1}=1,a_{0}=4$, for class (i) we used $a_{1}=1,a_{0}=7$, for class (iv) we used $a_{1}=4,a_{0}=1$, and finally for class (v) we used $a_{1}=7,a_{0}=1$. The data for “spherical”, originally due to [21], consist of 50,000 (x,y) pairs. Here, *Y* is again binary with p(y=0)=p(y=1)=1/2, but *X* is highly non-uniformly distributed over ≈4000 values. Again we classified these values as y–neutral or heavily/moderately biased toward or against y=0 and used this classification to choose values of $a_{y}$ accordingly.

## Data Availability

Not applicable.

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
