# Peer review of "On Generalized Schürmann Entropy Estimators"

_entropy, 2022, doi:10.3390/e24050680_

Round 1

Reviewer 1 Report

The author proposes new estimators of Shannon entropy by generalizing previous results proposed Schuermann and himself. The method consists in an ansatz for the average of z_i ln(z_i) as n_i \phi(n_i), where z_i = p_i N and p_i are the unknown probabilities needed to calculate the Shannon entropy. The choice of \phi is determinant to the accuracy of the approximation and previous works used functions based on the digamma function plus a definite integral from 0 to 1. The author first considers a generalization to a one-parameter class of functions where the integral runs from 0 to a and then further generalizes it making the parameter a box-dependent, a -> a_i. 

The proposed ansatz has an exact solution with a_i = (1-p_i)/p_i, which of course is not available, since the p_i are unknown. Moreover, the variance of the estimator usually blows up near the optimal choice of the a_i. 

I think this is a very interesting contribution, with great insights and simple numerical examples that illustrate the usefulness of the new approach and I recommend publication. One point that was not clear to me was the range of the parameters a_i. In the original proposal a_i=1 (eq.(17)) but the optimal solution (eq.(28) leads to very large a's (therefore very far from the original proposal), where the variance becomes large. The there is the message that a_i < O(1). So how is the optimal solution compatible with this message? Maybe the author could comment a little more on that.

Author Response

please check the response in the attachment.

Reviewer 2 Report

The paper investigates the problem of entropy estimation from finite samples, which is of wide interest for applications and is known to present some bias issues. The author generalises a previous result from Schürmann, which was an improvement on a previous paper of the author himself.

The content of the paper is of great interest and represents a big step forward, at least from a theoretical point of view. The application of the results is not so straightforward since, as clearly pointed out in the paper, it would require the knowledge of usually unknown quantities (the real pi's). Nevertheless, some useful considerations for practical use are presented.

1) The presentation though is very dense and difficult to follow. I ask that some steps be made clearer and some further explanation be provided, starting from the beginning of page 3 up to Eq. (27).

2) I also recommend the splitting of the article into sections (an idea would be one section as Introduction/Background, one for the improved estimator, one for the applications and a concluding one).

In addition to the observations 1) and 2), I point out the following.

p. 2: "Furthermore, since zE1(2z) is positive definite"; do you simply mean that it's a positive term? The expression "positive definite" is somewhat confusing, as it has its own specific meaning which is not the case here.

p. 3: In the second line of Eq. (20) it seems a factor n is missing

p. 3: In Eq. (20) how do you obtain the fourth line from the expression in the third one? Writing the E1's in integral form does not provide any obvious answer. Is there a substitution involved? What is that?

p. 3: After Eq. (21), I don't understand what follows. i) What is the "optimality" one wants to achieve? A zero bias? ii) How does Eq. (22) follow if the term zE1((1+a)z) is positive? iii) How do you show that E1(bz) < e-bz? Please provide sufficient explanation to the reader.

p. 3: In the second line of Eq. (23) it seems a factor n is missing.

p. 3: In the l.h.s. of Eq. (26) remove an open parenthesis.

p. 3: One verifies that Hopt in Eq. (27) has zero bias, by choosing the ai's that reduce the integration range in Eq. (26) to zero, but it's not immediately obvious to me without writing things down. Please make it clearer (here or before). Furthermore, is the "correction term whose bias vanishes..." the term nGni(ai) in the sum? Please clarify the presentation, it's too elliptic.

p. 3: In Eq. (30) if I'm not wrong it is gn+1(a) = gn(a) - (-a)n / n, not +; please check.

p. 4: In the caption of Fig. 2, why is a1* = 2.5 a bias-free value? Shouldn't it be a1* = 3 (from Eq. (28))?

p. 4: What is meant by aini < O(1)? Please clarify with words or use an unambiguous notation.

p. 4: In the paragraph starting with "Finally, we apply our estimator...", there is a missing parenthesis (after citation of [22]) and a wrong one after "= 1/2".

p. 4: results in Fig. 3 are said to agree perfectly for large N with those in [22], which the reader with great probability does not have access to; is there a way to reproduce them in your plot asking for the data or to reproduce the original plot, asking for editorial permission?

p. 5: Since it is important for any applications, it would be important to clarify in the conclusions what "judicious parameter choices" would be like. Can keeping all aini small (how small?) be always sufficient? Is that always possible?

In conclusion, I recommend the publication, but some work to clarify the too cryptic presentation is needed.

Author Response

please check the response in the attachment

Round 2

Reviewer 2 Report

Many points were made clearer and the readability of the paper has increased, but some points are still unclear to me.

(minor) p. 3, l. 1: "positive" is spelled wrongly

p. 3: I asked how can be shown that E1(bz) < e-bz for b>0 but no explanation to this was given. In fact, this does not seem to hold true for all b>0. Taking z=1 and, for example, b=1/4, numerical integration shows that E1(1/4) is actually larger than e-1/4. The inequality may hold for all large enough b, but this has to be shown. Please be more precise on this point and provide sufficient explanation to the reader.

p. 4: I had already pointed out this in the previous review but got no answer. In the caption of Fig. 2, why is a1* = 2.5 indicated as a bias-free value? Shouldn't it be a1* = 3 (from Eq. (28))? Some changes were made in the figure caption, but what I don't understand is why a1 and a2 are linked in such a way that, corresponding to the bias-free value a2*=7, it's a1=2.5 and not a1=a1*=3, the bias-free value. Also the a1 range in the picture stops at 2.5 and it's not clear why. It would be interesting and the most natural thing to see what happens at a1=3 with e.g. a2=1+3(a1-1) (so that a2(a1*)=a2*), or provide sufficient explanation why it's not.

Author Response

Thanks for reviewer's 2nd round comments, here is the response: 

1) I changed ``ositive" to ``positive"

2) I corrected the wrong inequality after Eq.(22) from   E_1(z)< exp(-z)  to

E_1(z)< exp(-z)/z. This has no further consequences.

3) I now give a more correct description of Fig. and a more complete explanation for the choice of parameters in it:

``a-values near the bias-optimal ones gave
estimates with zero almost zero bias and acceptable vari-
ance for the most undersampled case N = 2. Again,
using the the exact bias-optimal values would have given
unacceptably large variances for large N ."